# Towards a Long-Read Sequencing Approach for the Molecular Diagnosis of RPGR^ORF15^ Genetic Variants

**DOI:** 10.3390/ijms242316881

**Published:** 2023-11-28

**Authors:** Gabriele Bonetti, William Cozza, Andrea Bernini, Jurgen Kaftalli, Chiara Mareso, Francesca Cristofoli, Maria Chiara Medori, Leonardo Colombo, Salvatore Martella, Giovanni Staurenghi, Anna Paola Salvetti, Benedetto Falsini, Giorgio Placidi, Marcella Attanasio, Grazia Pertile, Mario Bengala, Francesca Bosello, Antonio Petracca, Fabiana D’Esposito, Benedetta Toschi, Paolo Lanzetta, Federico Ricci, Francesco Viola, Giuseppe Marceddu, Matteo Bertelli

**Affiliations:** 1MAGI’s LAB, 38068 Rovereto, Italy; chiara.medori@assomagi.org (M.C.M.); matteo.bertelli@assomagi.org (M.B.); 2Department of Pharmaceutical Sciences, University of Perugia, 06123 Perugia, Italy; 3MAGI Euregio, 39100 Bolzano, Italy; amministrazione@assomagi.org (W.C.); jurgen.kaftalli@assomagi.org (J.K.); chiara.mareso@assomagi.org (C.M.); f.desposito@imperial.ac.uk (F.D.); giuseppe.marceddu@assomagi.org (G.M.); 4Department of Biotechnology, Chemistry and Pharmacy, University of Siena, 53100 Siena, Italy; andrea.bernini@unisi.it; 5Department of Ophthalmology, ASST Santi Paolo e Carlo Hospital, University of Milan, 20142 Milan, Italy; leonardo.colombo.82@gmail.com (L.C.); martellasalvatore7@gmail.com (S.M.); 6Eye Clinic, Department of Biomedical and Clinical Science, Luigi Sacco Hospital, University of Milan, 20157 Milan, Italy; giovanni.staurenghi@unimi.it (G.S.); paola.anna.salvetti@gmail.com (A.P.S.); 7UOC Oculistica, Fondazione Policlinico Universitario “A. Gemelli” IRCCS, Largo Gemelli 8, 00168 Rome, Italygiorgio.placidi@guest.policlinicogemelli.it (G.P.); 8Istituto di Oftalmologia, Università Cattolica del Sacro Cuore, Largo Francesco Vito 1, 00168 Rome, Italy; 9Ospedale Sacrocuore Don Calabria, Viale Luigi Rizzardi, 4, 37024 Negrar di Valpolicella, Italy; marcella.attanasio@sacrocuore.it (M.A.); grazia.pertile@sacrocuore.it (G.P.); 10Medical Genetics Unit, Department of Oncohematology, Policlinico Tor Vergata, 00133 Rome, Italy; mario.bengala@ptvonline.it; 11Department of Surgical Sciences, Dentistry, Paediatrics and Gynaecology, Section of Ophthalmology, University of Verona, 37134 Verona, Italy; francesca.bosello87@gmail.com; 12Division of Medical Genetics, Fondazione IRCCS-Casa Sollievo della Sofferenza, 71013 San Giovanni Rotondo, Italy; a.petracca@operapadrepio.it; 13Imperial College Ophthalmic Research Group (ICORG) Unit, Imperial College, London NW1 5QH, UK; 14Eye Clinic, Department of Neurosciences, Reproductive Sciences and Dentistry, University of Naples Federico II, 80138 Naples, Italy; 15Section of Medical Genetics, Department of Medical and Oncological Area, University Hospital of Pisa, 56126 Pisa, Italy; b.toschi@ao-pisa.toscana.it; 16Department of Medicine-Ophthalmology, University of Udine, 33100 Udine, Italy; paolo.lanzetta@uniud.it; 17Istituto Europeo di Microchirurgia Oculare (IEMO), 33100 Udine, Italy; 18Department of Experimental Medicine, Tor Vergata University of Rome, Viale Oxford, 00133 Rome, Italy; rccfrc00@gmail.com; 19Department of Ophthalmology, Fondazione IRCCS Cà Granda, Clinica Regina Elena, 20122 Milan, Italy; francesco.viola@unimi.it; 20MAGISNAT, Atlanta Tech Park, 107 Technology Parkway, Peachtree Corners, GA 30092, USA

**Keywords:** long-read sequencing, RPGR, PacBio, molecular dynamics, molecular docking, TTLL5

## Abstract

Sequencing of the low-complexity ORF15 exon of RPGR, a gene correlated with retinitis pigmentosa and cone dystrophy, is difficult to achieve with NGS and Sanger sequencing. False results could lead to the inaccurate annotation of genetic variants in dbSNP and ClinVar databases, tools on which HGMD and Ensembl rely, finally resulting in incorrect genetic variants interpretation. This paper aims to propose PacBio sequencing as a feasible method to correctly detect genetic variants in low-complexity regions, such as the ORF15 exon of RPGR, and interpret their pathogenicity by structural studies. Biological samples from 75 patients affected by retinitis pigmentosa or cone dystrophy were analyzed with NGS and repeated with PacBio. The results showed that NGS has a low coverage of the ORF15 region, while PacBio was able to sequence the region of interest and detect eight genetic variants, of which four are likely pathogenic. Furthermore, molecular modeling and dynamics of the RPGR Glu-Gly repeats binding to TTLL5 allowed for the structural evaluation of the variants, providing a way to predict their pathogenicity. Therefore, we propose PacBio sequencing as a standard procedure in diagnostic research for sequencing low-complexity regions such as RPGR^ORF15^, aiding in the correct annotation of genetic variants in online databases.

## 1. Introduction

Next-generation sequencing (NGS) was a breakthrough in molecular biology, providing a cheaper and faster method for sequencing DNA compared to the previous techniques, such as Sanger sequencing [1]. NGS techniques, among which Illumina is the most represented, rely on specific laboratory procedures, specifically DNA fragmentation, DNA end-repair, adapter ligation, surface attachment, and in situ amplification. A huge amount of data is produced, and thanks to bioinformatics analysis, it is possible to detect genetic mutations like single-nucleotide polymorphisms (SNPs) and indels [1,2]. NGS is considered a “short-reading” approach because it permits the sequencing of short fragments of DNA of 300 bp length (on average), this being its main limitation because it impedes the sequencing of low-complexity and/or GC-rich regions [3]. NGS strongly supports molecular genetics, allowing us to identify many genetic variants and to define the genetic onset of many genetic diseases. However, its limitations impede the study of many genes, which present low-complexity regions, and of specific chromosomal regions, in which massive arrays of tandem repeats predominate [3]. These limitations require developing a new sequencing technique: Third-generation sequencing (TGS). TGS, unlike NGS, comprises long-read sequencing approaches and is represented mainly by Pacific Biotechnology (PacBio) or Oxford Nanopore [2,3].

PacBio works on circular DNA fragments, while Oxford Nanopore works on linear DNA fragments. Moreover, PacBio relies on fluorescently labeled nucleotides for nucleotide detection, while Oxford Nanopore relies on an electronic signal disruption caused by the passage of DNA in a nanopore. The PacBio technique can analyze sequences up to 300 Kb, while Oxford Nanopore can analyze sequences up to 4 Mb. Finally, they can both detect epigenetic modifications. These techniques also have some disadvantages. PacBio is expensive, both in the instrumentation and in the sequencing cost. Moreover, the accuracy of these techniques was originally low, and it did not permit their scalability in a diagnostic setting. PacBio accuracy has recently increased dramatically, reaching 99.99%, while Oxford Nanopore recently reached 99.9% [2,3,4].

This work aims to propose PacBio sequencing for diagnostic applications, focusing on *RPGR^ORF15^* sequencing. *RPGR* is a gene related to retinitis pigmentosa (RP) and cone dystrophy. Mutations to *RPGR* are responsible for over 70% of the X-linked Retinitis pigmentosa cases and for over 73% of all cone and rode cone dystrophy cases [5,6]. *RPGR* has two isoforms, namely *RPGR^default^* and *RPGR^ORF15^*. *RPGR^default^* comprises 19 exons, while *RPGR^ORF15^* shares the first 14 exons with *RPGR^default^*, along with a distinct ORF15 exon [7], [8]. The first ten exons code for an RCC1 (regulator of chromatic condensation 1-like) domain, which is involved in the regulation of small GTPases, while the ORF15 exon is a 1 kb-long, highly repetitive, low complexity, purine-rich region, terminating with a C-terminal tail region with unknown function (basic domain) [7,9]. The low complexity of ORF15 makes NGS or Sanger sequencing ineffective for routine sequencing, thus increasing the possibility of false negative or false positive results [8,10]. Considering that ORF15 is a mutation hotspot of *RPGR* (Appendix A), a great number of variants associated with this region are uploaded in the database of single nucleotide variants (dbSNP), commonly used worldwide to detect variants and diagnose hereditary genetic diseases and on which online databases and tools such as HGMD and Ensembl rely [11,12]. Indeed, most of the genetic variants correlated to Retinitis Pigmentosa have been found in *RPGR^ORF15^* [13]; thus, a robust, accurate, and scalable test to sequence ORF15 is necessary for a precise genetic diagnosis. Finally, precise genetic variants identification will permit the implementation of personalized medicine strategies, like gene therapy, thus providing a new treatment option to RP patients [8].

Regarding its molecular structure, the photoreceptor-specific ORF15 variant of *RPGR* RPGR^ORF15^ contains multiple Glu-Gly tandem repeats and a C-terminal basic domain unknown in function and is localized to the connective cilium where it is thought to regulate cargo transport. The tubulin tyrosine ligase like-5 (TTLL5) glutamylates RPGR^ORF15^ in its Glu-Gly–rich repetitive region, which contains motifs homologous to the α-tubulin C-terminal tail; loss of glutamylation has pathological consequences in developing retinal dystrophy [7]. The C-terminal basic domain of RPGR^ORF15^ interacts with the TTLL5 noncatalytic cofactor interaction domain, which is unique among the glutamylases of the TTLL family and targets TTLL5 to glutamate RPGR as a result. TTLL5 is the only glutamylase in the TTLL family that interacts with RPGR^ORF15^ when expressed transiently in cells [7]. While the association of TTLL5 variants with loss of glutamylation due to enzymatic deficiency has already been characterized [14,15,16], less is known for the loss of glutamylation due to variants affecting the RPGR^ORF15^. The interaction of the TTLL enzyme family with regions rich in Glu-Gly repeats has been studied in detail for some cases as glutamylation of tubulin by TTLL4 and TTLL6 [17], and while the general mechanism of initiation and elongation can be extended to the whole family, specific characterization of the TTLL5/RPGR^ORF15^ interaction to support the effect of variants on the Glu-Gly-rich region is lacking.

## 2. Results

### 2.1. Study Cohort Characteristics

Table 1 reports the clinical characteristics of the probands analyzed for this study.

### 2.2. NGS Sequencing Coverage

Appendix A shows four examples of pitfalls in the NGS sequencing coverage of *RPGR^ORF15^* exon 15. Appendix A summarizes the NGS sequencing coverage of *RPGR* ORF15 exon in the analyzed samples. As can be seen from Figure 1, NGS sequencing coverage is low, and most of the samples (73 patients) had a sequencing coverage between 50.0% and 65.0%.

### 2.3. PacBio Sequencing Results

Table 2 reports the genetic variants identified in the *RPGR* gene with PacBio sequencing. On average, 98.12% of reads mapped to the entire genome. Within the region, coverage averaged 3587.44 with a standard deviation of 436.58. PacBio sequencing was able to detect in eight unrelated patients eight genetic variants that were not identified with NGS sequencing, four of which were predicted to be likely pathogenic. Table 3 compares the clinical characteristics between the probands in which *RPGR* variants were identified and the other probands.

### 2.4. Structural Analysis of TTLL5 Core Domain Binding

The RPGR^ORF15^ sequence diverges from the default variant, which is not glutamylated, at its C-terminal half, consisting of a region rich in Glu-Gly repeats followed by a C-terminal basic domain (Figure 2). The repeat region contains glutamate-rich motifs (GEEEG) homologous to the a-tubulin C-terminal tail and is glutamylated by the TTLL5 core domain. The basic domain, highly conserved among vertebrates, is crucial for recruiting TTLL5 [7,18].

The variants we identified are of the stop gained type, leading to the loss of almost the entire length of the ORF region and basic domain, except for the NM_001034853.2:c.2203_2226del p.(His735_Glu742del) (Figure 2). Such a variant leads to the loss of a GEGE tandem repeat but retains the structure of RPGR^ORF15^ and was therefore used as a model for the interaction. The model of the TTLL5 core domain was obtained by homology modeling to the most recent evidence of TTLL6 structure, where an initiation analog is bound to the active site. The analog mimics a di-Glu peptide where the donor glutamate is linked to the γ-carboxylic acid of acceptor glutamate through a phosphinate. The di-Glu was split by removing the phosphinate and was transformed into the donor and the acceptor glutamate. The latter was interpreted as Glu737 in RPGR^ORF15^, the glutamination site in the deleted peptide segment from the NM_001034853.2:c.2203_2226del p.(His735_Glu742del) variant. The Glu737 was then expanded at each terminus to reproduce the structure of the deleted peptide and then docked into the crevice of the active site by holding Glu737 fixed, working as an anchor. Such experiment-based positional restraint imposes the right orientation for the peptide to be docked and lowers its conformational space, thus improving the accuracy of the docked structures. The final model shows the peptide GEEEHGE737GEEEE filling the positively charged crevice centered on the active site with each glutamate sidechain closely interacting with at least one lysine/arginine residue (Figure 3). Interestingly, the only non-Glu/non-Gly residue, the His375, finds its sidechain deeply inserted in a sub pocket of the crevice and involved in a π-interaction with Ser307; such interactions have been recently proposed as relevant in protein-protein binding with stabilization energy ranging from −20 kJ mol^−1^ to −40 kJ mol^−1^ [19].

Molecular dynamics simulations of free and bound TTLL5 were performed to further characterize the binding of the RPGR^ORF15^ glutamate-rich segments. Comparing the average structure of the bound TTLL5 simulations with the experimental structure of TTLL6 bound to the initial analog (PDB entry 6VZW) showed that the β6-β7 loop adopts a different conformation (Figure 4b), in accordance with the results reported in [20] on the importance of this loop to the unique activity of TTLL5. Jointly with loops α1-β1 and α2-β3, it forms the crevice where the glutamate-rich segment can bind (Figure 3). Furthermore, the β6-β7 is also rearranged in the bound simulation, adapting to an open conformation to accommodate the RPGR^ORF15^ segments (Figure 4A), as also shown by the center of mass (COM) distance between the respective loops forming the crevice (Appendix A).

To understand how the binding of different glutamate-rich segments would affect the active site of TTLL5, we performed the modeling and molecular dynamics simulations of two more peptide structures of the same length of the Glu737 centered GEEEHGE737GEEEE: (i) the peptide resulting from the NM_001034853.2:c.2203_2226del p.(His735_Glu742del) variant, centered on E734, EEGGEEE734GDREE; (ii) a peptide reproducing different glutamylation site, centered on E870, EGEGEEE870GEEGE. To assess the structural stability of TTLL5 core domain bound to such different segments, the RMSD of each bound peptide was evaluated. The results showed the two wildtype segments GEEEHGE737GEEEG and EGEGEEE870GEEGE to be stable, while the post-deletion sequence EEGGEEE734GDREE showed significant deviation, as shown in Figure 5.

## 3. Discussion

Genetic mutations to *RPGR* are responsible for over 70% of X-linked Retinitis pigmentosa and rode cone dystrophy cases [5,6,21]. Nevertheless, sequencing of *RPGR^ORF15^*, one of *RPGR*’s two main isoforms, is still challenging with NGS, and it can be associated with a considerable amount of false results [8,10]. TGS has recently been proposed for diagnostic purposes, considering that its accuracy has highly improved in recent years [22,23]. Thus, this study aimed to propose PacBio for diagnostic sequencing, focusing on *RPGR^ORF15^*. Furthermore, molecular modeling and dynamics of the RPGR GLU-GLY repeats binding to the TTLL5 core domain allowed for the structural evaluation of the variants, providing a way to predict their pathogenicity.

As shown in Figure 1 and Appendix A, NGS is not suitable for sequencing *RPGR^ORF15^* because its coverage highly decreases in exon 15, resulting in possible false results. On the other hand, in our study, PacBio sequencing identified eight genetic variants of *RPGR^ORF15^* that were not detected by NGS sequencing in the analyzed patients (Table 2). Among them, four were predicted to be likely pathogenic. In addition, the comparison of the clinical data of the patients with *RPGR^ORF15^* mutations with the clinical data of the other analyzed patients revealed that the age and the age of onset were significantly different (*p* < 0.01), suggesting that *RPGR^ORF15^* mutations could lead to an earlier onset pathology (Table 3). These results support the use of PacBio sequencing for genetic diagnosis of *RPGR^ORF15^* mutation, considering that NGS did not identify many genetic variants and that these variants seem to be involved in retinal dystrophy onset. NGS of the low-complexity region can also provide false results, which in turn could lead to the inaccurate annotation of genetic variants in dbSNP [24] and ClinVar [25] databases, tools on which HGMD [11] and Ensembl [12] rely, finally resulting in incorrect genetic variant interpretation and impeding an efficient genetic diagnosis.

Considering the occurrence of pathogenic variants in the unstructured ORF15 domain, we modeled the interaction between the RPGR^ORF15^ Glu-Gly rich segments and the TTLL5 core domain to derive structural information about the interaction with the observed variants. Indeed, the results showed that Glu-Gly rich segments of RPGR^ORF15^ similarly bind TTLL5 to that observed in the α-tubulin tail, with loops α1-β1, α2-β3, and β6-β7 adapting to an open conformation and stabilizing the bound segment by forming electrostatic interactions with the negatively charged glutamates.

The main limitation of this study is the small number of patients analyzed with PacBio sequencing. A bigger cohort of patients could support the main aim of the article, proving that TGS is the best solution for sequencing low-complexity regions and that NGS could result in false negative or false positive results. Moreover, further molecular modeling studies on the uncharacterized interaction between RPGR^ORF15^ basic domain and the cofactor interaction domain of TTL5 could yield insight into the pathogenicity of the variants found in this region. Finally, more detailed clinical data could be useful in finding possible correlations between genetic variants in *RPGR^ORF15^* and disease severity. Nevertheless, long-read sequencing could be a feasible approach to sequence low-complexity regions for diagnostic purposes, considering NGS as a first option for the other regions. At the same time, molecular dynamics can add useful knowledge of proteins involved in disease onset, increasing our chance to identify pathogenic variants, and searching for possible new therapeutic targets.

## 4. Materials and Methods

### 4.1. Subjects and Samples

We analyzed 75 Caucasian subjects diagnosed with retinitis pigmentosa or cone dystrophy. All patients were recruited and underwent detailed clinical examinations by different Italian hospitals. All patients underwent pre-test counseling, during which clinical data—including personal and family history—were collected and evaluated. The patients were informed about the significance of genetic testing. All of them gave their written informed consent in compliance with the Declaration of Helsinki. Genomic DNA was isolated from peripheral blood or saliva using a commercial kit (SaMag Blood DNA Extraction Kit (Sacace Biotechnologies, Como, Italy)) according to the manufacturer’s instructions.

### 4.2. NGS Sequencing

A large custom panel [approximately 2.4 Mb cumulative target length (GRCh38/hg38)], encompassing several genes related to retinitis pigmentosa and cone dystrophy among which *RPGR*, were used for NGS analysis. The DNA probe set was designed to capture the coding exons and flanking regions of each gene of the panels using the Twist Custom Panel Design Technology (Twist Bioscience, South San Francisco, CA, USA https://www.twistbioscience.com/products/ngs accessed on 20 September 2023). The subpanel of analyzed genes were selected on the basis of literature or databases [Human Gene Mutation Database (HGMD Professional), Online Mendelian Inheritance in Man (OMIM), Orphanet, NCBI GeneReviews, NCBI PubMed and specific database].

Library preparation from genomic DNA samples was performed according to the manufacturer’s protocol using the Twist Library Preparation EF Kit and Twist Universal Adapter (UDI) System with Standard Hybridization Target Enrichment (Twist Bioscience). Briefly for library preparation, 50 ng of each genomic DNA was enzymatically fragmented to yield fragments of 450–550 bp, and end repaired and dA-Tailed in the same reaction. Then, the Twist universal adaptor was ligated on fragments, SPRI purified and enriched by 7 PCR cycles with Twist Unique Dual Index Primers. Next, 187.5 ng of each purified Libraries was then hybridized to the Twist oligo probe capture library for 16 hr in a twelve-plex reaction. After hybridization, washing, and elution, the eluted fraction was PCR-amplified with 9 cycles and purified. A 150 bp paired-end reads sequencing was performed on MiSeq personal sequencer (Illumina, San Diego, CA, USA) according to the manufacturer’s instructions. A total of 24 pool library samples were loaded on Miseq using MiSeq V3 kit.

### 4.3. SMRTbell Library Preparation and TGS Sequencing

The SMRTbell library to sequence *RPGR^ORF15^* was prepared following the PacBio protocol guidelines Procedure & Checklist—Preparing SMRTbell^®^ Libraries using PacBio^®^ Barcoded Universal Primers for Multiplexing Amplicons—Part Number 101–791-800 Version 02 (April 2020) [26]. All assays were performed as already described in our publication [27].

For the first-round PCR, forward target-specific primer (5′-GACAG-TTACATGGAAGGTGCAA-3′) and reverse target-specific primer (5′-TACCAG-TGCCTCCTATTGTCTT-3′) tailed to F/R universal sequences were designed to amplify the ~1.7 kb DNA fragment spanning the entire *RPGR^ORF15^* gene exon 15 sequence (NM_001034853). Primer design was performed using the freely available program Primer3web [28] and, to avoid mispriming, the primer couple was tested in silico using NCBI PrimerBLAST [29] and the UCSC tool In-Silico PCR [30]. First- and second-round PCR were performed using PrimeSTAR GXL Polymerase (TaKaRa Bio, Shiga, Japan). The polymerase was tested and confirmed in our precedent work. Second round PCR products were purified using 0.45X AMPure PB beads (Pacific Biosciences, Menlo Park, CA, USA) and DNA concentrations were read on a Qubit 2.0 Fluorometer using the Qubit dsDNA Broad Range Assay Kit (Invitrogen Life Technologies, Carlsbad, CA, USA).

For an amplicon size ranging from 1 to 3 kb, the input DNA amounts per pool should be between 500–1000 ng. Depending on the recorded concentrations, we proceeded by calculating the necessary microliters of each sample so as to obtain a total amount of 1000 ng. The pooled PCR amplicons were then concentrated using AMPure PB Beads. The next step consisted in the construction of the SMRTbell library, which involves DNA damage repair, End-repair/A-tailing, and adapter ligation. All the steps are described in the PacBio procedure with specific reagent and amounts [26]. The SMRTbell Templates were then purified again with AMPure PB before sequencing.

### 4.4. Bioinformatics and Genetic Variants Classification

The pathogenicity of all identified variants was evaluated according to the American College of Medical Genetics and Genomics guidelines (ACMG) [31], with the help of VarSome [32], dbSNP [24], ClinVar [25], and gnomAD [33] databases, thanks to an in-house bioinformatics pipeline [34]. Fastq (forward-reverse) files were obtained after sequencing. Bioinformatic analysis was performed as previously described [35,36]. Briefly, the sequencing reads were mapped to the reference genome (hg38/GRCh38) using Burrow-Wheeler Aligner (version 0.7.17-r1188) software. Duplicates were removed using SAMBAMBA (version 0.6.7) and MarkDuplicates GATK (version 4.0.0.0). The BAM alignment files generated were refined by local realignment and base quality score recalibration, using the RealignerTargetCreator and IndelRealigner GATK tools. Minor allele frequencies (MAF) were retrieved from the Genome Aggregation Database [33]. Long-read sequencing data were analyzed using an in-house bioinformatics pipeline [37].

### 4.5. Molecular Modeling of TTLL5-RPGR^ORF15^ Interactions

The model of the TTLL5 core domain was obtained by homology modeling to the TTLL6 structure reported in PDB entry 6VZW, where an initiation analog is bound to the active site of TTLL6. The analog mimics a di-Glu peptide where the donor glutamate is linked to γ-carboxylic acid of acceptor glutamate through a phosphinate (donor and acceptor refer to the free glutamate to be linked and the receiving glutamate, respectively). The main chain residues flanking the di-Glu are replaced by ethylamine at C-term and acetate at N-term. The intermediate’s structure and binding geometry were then retained in the modeled TTLL5 and exploited to generate the enzyme-bound form of RPGR^ORF15^. The di-Glu was split by removing the phosphinate and was transformed into the donor and the acceptor glutamate. The latter was interpreted as Glu737 in RPGR^ORF15^, and after ethylamine and acetate capping removal, the Glu737 was expanded at each terminus to reproduce the sequence of the deleted peptide plus the flanking amino acids to the extent necessary to cover the active site crevice, leading to the final 12-mer GEEEHGE737GEEEE peptide. The structure of the expanded peptide was docked to the protein structure by holding Glu737 fixed, acting as a covalent anchor to the upstream and downstream segments. The process was repeated for two other peptides, namely EEGGEEE734GDREE and EGEGEEE870GEEGE.

Docking was performed using Autodock Vina 1.2 [38] and MAGI-Dock [39], a PyMol plugin that generates the docking boxes [37] around the active site crevice. Two separate docking runs were performed, one for each of the upstream and downstream segments of Glu737.

Finally, the resulting conformation was energetically refined by molecular dynamics simulations. We used CHARMM-GUI [40] to generate the solvated system and GROMACS [41] to run the simulation. The segment’s flanking residues were neutralized to prevent unwanted interaction, and the resulting docked structure was used as the simulation starting structure. The protein complex was placed in a triclinic box with a minimum 1.2 nm spacing on each side, and the system was solvated using TIP3P water molecules and neutralized with K^+^/Cl^−^. Sequential minimization and position-restrained equilibrations in the NVT and NPT ensemble were performed before a 200 ns long production run. Root Mean Squared Deviation (RMSD) and trajectory clustering were analyzed using the GROMACS tools gmx rms and gmx cluster, respectively.

## Figures and Tables

**Figure 1 ijms-24-16881-f001:**
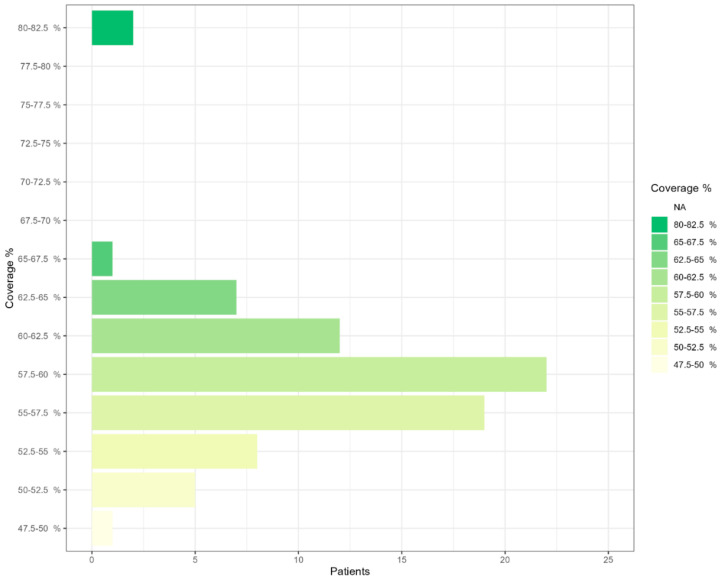
NGS sequencing coverage of RPGR ORF15 exon in the analyzed samples.

**Figure 2 ijms-24-16881-f002:**
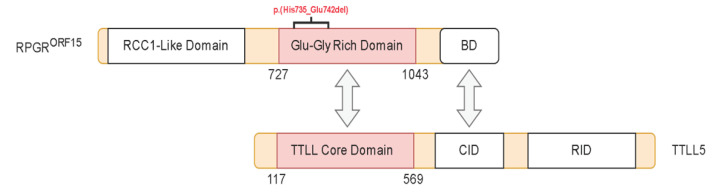
Graphical representation of RPGR^ORF15^ and TTLL5 interaction. The basic domain (BD) of RPGR^ORF15^ recruits TTLL5 and binds to its noncatalytic CID domain. The in-frame deletion is indicated in red.

**Figure 3 ijms-24-16881-f003:**
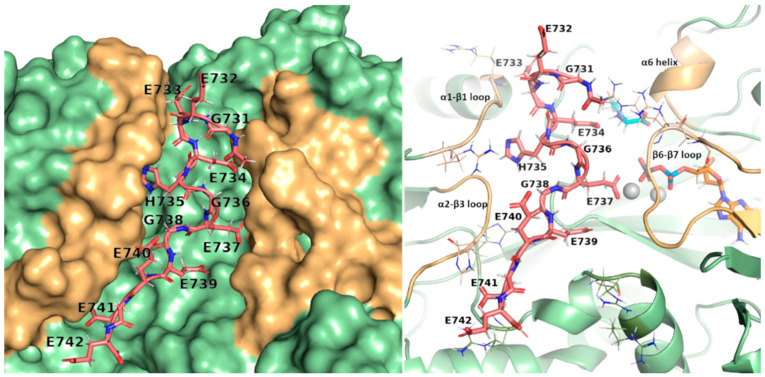
Surface and cartoon view of TTLL5 active site. The peptide (in pink, shown as sticks) fits into the crevice centered at the active site, with Glu737 directed towards ADP. Glutamates align to form electrostatic interactions with the positively charged residues (shown as lines in the cartoon view), stabilizing the peptide into the pocket. The structural components surrounding the peptide chain are shown in light orange (loops α1-β1, α2-β3, β6-β7, and helix α6), with the rest of the protein shown in green; the donor glutamate is shown in cyan; magnesium ions and ADP are shown in gray and orange, respectively.

**Figure 4 ijms-24-16881-f004:**
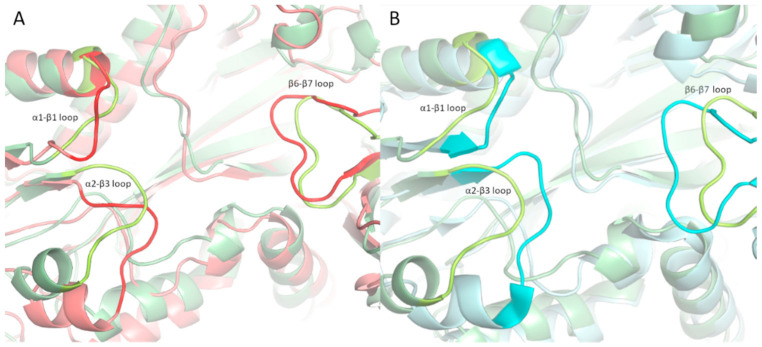
Loop configurations around the binding site. (**A**) Comparison of the binding cavity between the average structure of the bound (light green) and free (light red) TTLL5. The β6-β7 loop is slightly displaced to accommodate the peptide chain. (**B**) Comparison of the binding cavity between the average structure of the TTLL5 bound to the GEEEHGE737GEEEE (light green) and TTLL6 (light cyan). The open configuration is evident in TTLL5, with a notable displacement of the β6-β7 loop. Loops are marked with higher opacity.

**Figure 5 ijms-24-16881-f005:**
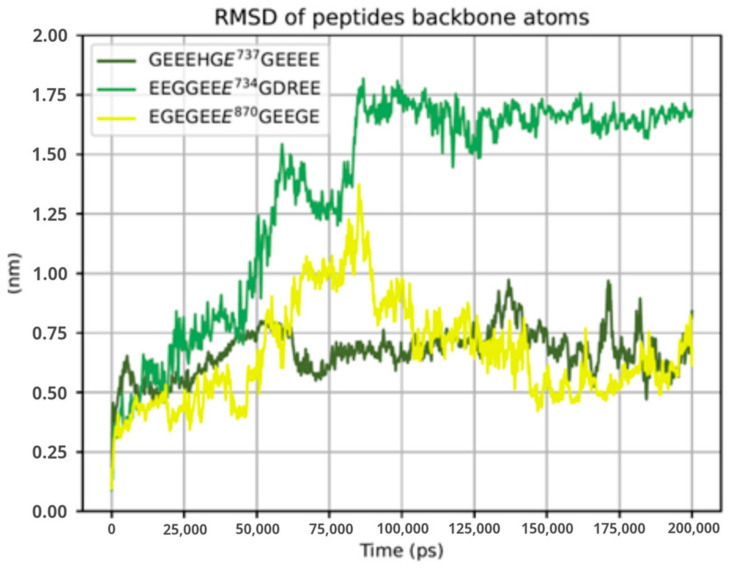
RMSD of each peptide chain bound to TTLL5. While the variant (shown in light green) is characterized by large movements for the first 100 ns, the two wildtype segments (shown in dark green and yellow) are subject to fewer fluctuations, especially the one with Glu737 as the glutamylation site.

**Table 1 ijms-24-16881-t001:** Clinical data of the patients. The pathology was considered familiar if other cases were present in their clinical familiar history. All the analyzed patients were unrelated.

Characteristics		Case Subjects (n = 75)
**Age (years)**	Mean	51 ± 17
	Median	52 ± 17
**Females/Males**		35/40 (47%/53%)
**Diagnosis**	Hereditary non-syndromic retinal dystrophies	5 (7%)
	Retinitis pigmentosa	56 (75%)
	Cone dystrophy	8 (10%)
	Macular distrophy	6 (8%)
**Age of onset (years)**	Mean	25 ± 16
	Median	24 ± 16
	Unknown	n = 8
**Familiarity**	Familiar	26 (35%)
	Sporadic	40 (53%)
	Unknown	9 (12%)

**Table 2 ijms-24-16881-t002:** Genetic variants identified in RPGR gene in patients with PacBio sequencing.

ID	Nucleotide Variant	rsID	Verdict	Zygosity
1	NM_001034853.2:c.2919_2940dup	/	Likely Pathogenic	0/1
2	NM_001034853.2:c.2203_2226del	rs768423834	Uncertain Significance	0/1
3	NM_001034853.2:c.2203_2226del	rs768423834	Uncertain Significance	1/1
4	NM_001034853.2:c.2820_2841dup	/	Likely Pathogenic	0/1
5	NM_001034853:c.1871A>C	/	Uncertain Significance	0/1
6	NM_001034853:c.3423G>T	/	Uncertain Significance	1/1
7	NM_001034853.2:c.3262_3263insA	/	Likely Pathogenic	0/1
8	NM_001034853.2:c.2820_2841dup	/	Likely Pathogenic	0/1

**Table 3 ijms-24-16881-t003:** Clinical characteristics of the probands in which *RPGR* variants were identified and of the other probands. The pathology was considered familiar if other cases were present in their clinical familiar history. All the analyzed patients were unrelated.

Characteristics		Patients with *RPGR^ORF15^* Mutation (n = 8)	Case Subjects (n = 67)	*p*-Value
**Age (years)**	Mean	35 ± 11	53 ± 15	<0.05
	Median	36 ± 11	53 ± 15	
**Females/Males**		4/4 (50%/50%)	31/36 (46%/54%)	-
**Diagnosis**	Hereditary non-syndromic retinal dystrophies	0 (0%)	5 (7%)	-
	Retinitis pigmentosa	3 (37.5%)	53 (79%)	
	Cone dystrophy	2 (25%)	6 (9%)	
	Macular dystrophy	3 (37.5%)	3 (5%)	
**Age of onset (years)**	Mean	13 ±11	27 ± 16	<0.05
	Median	10 ± 11	27 ± 16	
	Unknown	n = 0	n = 8	
**Familiarity**	Familiar	2 (25%)	24 (36%)	-
	Sporadic	3 (37.5%)	37 (55%)	
	Unknown	3 (37.5%)	6 (9%)	

## Data Availability

All data are contained in the manuscript and in the Appendix A.

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
