# Peer review of "Towards a Long-Read Sequencing Approach for the Molecular Diagnosis of RPGRORF15 Genetic Variants"

_ijms, 2023, doi:10.3390/ijms242316881_

Round 1

Reviewer 1 Report

Comments and Suggestions for Authors

In the present manuscript, it indicated that PacBio sequencing improve the accuracy in the low-complexity regions such as the ORF15 exon of RPGR and proposed PacBio sequencing as a standard procedure in diagnostic research for low-complexity regions. The molecular modeling indicated that the structure in variants is predicted as the pathogenic. However, the manuscript is not well-written. I recommend that this paper accepted after minor revision.

1. Although the coverage of NGS sequencing is presented, the coverage of PacBio sequencing is not presented.

2. It should clarify that the coverage (depth) both NGS and PacBio sequencing that were used for variant calling.

3. In NGS sequencing on Materials and Methods, the explanation is insufficient. Was the DNA captured using the Twist Custom Panel kit? Was DNA sequenced using MiSeq according to the manufacturer’s protocol on NovaSeq 6000 platform?

4. It should clarify that the software for mapping, variant calling.

5. It is difficult to understand the Figure S1 and Figure S2 due to the small figure size and the poor explanation of figures.

Author Response

In the present manuscript, it indicated that PacBio sequencing improve the accuracy in the low-complexity regions such as the ORF15 exon of RPGR and proposed PacBio sequencing as a standard procedure in diagnostic research for low-complexity regions. The molecular modeling indicated that the structure in variants is predicted as the pathogenic. However, the manuscript is not well-written. I recommend that this paper accepted after minor revision.

  1. Although the coverage of NGS sequencing is presented, the coverage of PacBio sequencing is not presented.

Considering the coverage of PacBio, we wrote at lines 154,155: “On average 98,12% of reads mapped to the entire genome. Within the region, coverage averaged 3587.44 with a standard deviation of 436.58.”

It should clarify that the coverage (depth) both NGS and PacBio sequencing that were used for variant calling.

Figure S2 represents NGS coverage of RPGRORF 15 exon 15. Only this region has a coverage lower than 10X. All the other regions were analyzed with NGS with a higher coverage than 10X (60X on average). This concept was added to the legend of Figure S2.

Considering the coverage of PacBio, we wrote at lines 154,155: “On average 98,12% of reads mapped to the entire genome. Within the region, coverage averaged 3587.44 with a standard deviation of 436.58.”

  1. In NGS sequencing on Materials and Methods, the explanation is insufficient. Was the DNA captured using the Twist Custom Panel kit? Was DNA sequenced using MiSeq according to the manufacturer’s protocol on NovaSeq 6000 platform?

Dear, in lines 307-335 we added all the relevant information:

“A large custom panel [approximately 2.4 Mb cumulative target length (GRCh38/hg38)], encompassing several genes related to retinitis pigmentosa and cone dystrophy among which RPGR, were used for NGS analysis. The DNA probe set was designed to capture the coding exons and flanking regions of each gene of the panels using the Twist Custom Panel Design Technology (Twist Biosci-ence, https://www.twistbioscience.com/products/ngs). The subpanel of analyzed genes were selected on the basis of literature or databases [Human Gene Mutation Database (HGMD Professional), Online Mendelian Inheritance in Man (OMIM), Orphanet, NCBI GeneReviews, NCBI PubMed and specific database].

Library preparation from genomic DNA samples was performed according to the manufacturer’s protocol using the Twist Library Preparation EF Kit and Twist Universal Adapter (UDI) System with Standard Hybridization Target Enrichment (Twist Bioscience). Briefly for library preparation, 50 ng of each genomic DNA was enzymatically fragmented to yield fragments of 450-550 bp and end repaired and dA-Tailing in the same reaction. Then, the Twist universal adaptor was ligated on fragments, SPRI purified and enriched by 7 PCR cycles with Twist Unique Dual Index Primers. 187.5 ng of each purified Libraries was then hybridized to the Twist oligo probe capture library for 16 hr in a twelve-plex reaction. After hybridization, washing, and elution, the eluted fraction was PCR-amplified with 9 cycles and purified. A 150 bp paired-end reads sequencing was performed on MiSeq personal sequencer (Illumina, San Diego, CA) according to the manufacturer's instructions. 24 pool library samples were loaded on Miseq using MiSeq V3 kit.”

  1. It should clarify that the software for mapping, variant calling.

Dear, in lines 372-384 we added some relevant information and cited some of our relevant scientific articles published on the bioinformatic analysis of sequencing data (ref. 35,37,39):

“4.4. Bioinformatics and genetic variants classification

The pathogenicity of all identified variants was evaluated according to American College of Medical Genetics and Genomics guidelines (ACMG) [32], with the help of VarSome [33], dbSNP [25], ClinVar [26] and gnomAD [34] databases, thanks to an in-house bioinformatics pipeline [35]. Fastq (forward-reverse) files were obtained after sequencing. Bioinformatic analysis was performed as previously described [36,37]. Briefly, the sequencing reads were mapped to the reference genome (hg38/GRCh38) using Burrow-Wheeler Aligner (version 0.7.17-r1188) software. Duplicates were re-moved using SAMBAMBA (version 0.6.7) and MarkDuplicates GATK (version 4.0.0.0). The BAM alignment files generated were refined by local realignment and base quality score recalibration, using the RealignerTargetCreator and IndelRealigner GATK tools. Minor allele frequencies (MAF) were retrieved from the Genome Aggregation Data-base [38].  Long-read sequencing data were analyzed using an in-house bioinformat-ics pipeline [39].”

  1. It is difficult to understand the Figure S1 and Figure S2 due to the small figure size and the poor explanation of figures.

We increased the figures size and modified the legend. Now they should be more understandable.

Reviewer 2 Report

Comments and Suggestions for Authors

This study discovered a new pathogenic mutation in ORF15 exon domain of the RPGR gene, which plays an important function in Retinitis pigmentosa and core dystrophy. To identify the mutations, they suggest the Pacbio method rather than the existing NGS method (illumina).

There are many confusing parts, and also some parts lack the explanations. So, I comment a few questions.

1. The NGS experimental method (MiSeq or NovaSeq6000) is described too briefly. The author need more detailed experimental techniques (such as library construction, sample preparation, exome seq, target seq, or WGS, and analysis tools).

2. The overall manuscript describes the Illumina method as the NGS method, but PacBio is also one of the NGS methods which may mislead readers.

3. In Table 1, the total number of cases is 75, but the total number of Femail/male is 77. 

4. There are 26 family samples, and it is necessary to express whether or not they are family in Table 2 and Table 3.

5. The most important findings of this study appear to be p. (His735_Glu742del). However, this mutation was found in only two samples, and if the two samples belong to one family, the marker is a family-specific mutation, which is likely to be difficult to use in diagnose genetic diseases in the future.

6. Although the authors predicted protein structure for p.(His735_Glu742del) mutation, the Table 2 shows the muation was hetero in one sample and homozygotes in the other sample. If the symptoms of the two samples are almost the same, it is necessary to explain whether the Genetic mode of the Mutation is dominant mode or is likely to be influenced by other genes.

7. Supplementary material lacks description and makes it difficult to understand the table or picture.

Considering the current technology and NGS cost, it seems effective to first WGS with Illumina Sequence to find the causative factors at the Genome-wide level, and if you can't find it, it seems effective to sequence with PacBio the area that is difficult to test with Illumina among the candidate genes.

Author Response

This study discovered a new pathogenic mutation in ORF15 exon domain of the RPGR gene, which plays an important function in Retinitis pigmentosa and core dystrophy. To identify the mutations, they suggest the Pacbio method rather than the existing NGS method (illumina).

There are many confusing parts, and also some parts lack the explanations. So, I comment a few questions.

  1. The NGS experimental method (MiSeq or NovaSeq6000) is described too briefly. The author need more detailed experimental techniques (such as library construction, sample preparation, exome seq, target seq, or WGS, and analysis tools).

Dear, in lines 307-335 we added all the relevant information:

“A large custom panel [approximately 2.4 Mb cumulative target length (GRCh38/hg38)], encompassing several genes related to retinitis pigmentosa and cone dystrophy among which RPGR, were used for NGS analysis. The DNA probe set was designed to capture the coding exons and flanking regions of each gene of the panels using the Twist Custom Panel Design Technology (Twist Biosci-ence, https://www.twistbioscience.com/products/ngs). The subpanel of analyzed genes were selected on the basis of literature or databases [Human Gene Mutation Database (HGMD Professional), Online Mendelian Inheritance in Man (OMIM), Orphanet, NCBI GeneReviews, NCBI PubMed and specific database].

Library preparation from genomic DNA samples was performed according to the manufacturer’s protocol using the Twist Library Preparation EF Kit and Twist Universal Adapter (UDI) System with Standard Hybridization Target Enrichment (Twist Bioscience). Briefly for library preparation, 50 ng of each genomic DNA was enzymatically fragmented to yield fragments of 450-550 bp and end repaired and dA-Tailing in the same reaction. Then, the Twist universal adaptor was ligated on fragments, SPRI purified and enriched by 7 PCR cycles with Twist Unique Dual Index Primers. 187.5 ng of each purified Libraries was then hybridized to the Twist oligo probe capture library for 16 hr in a twelve-plex reaction. After hybridization, washing, and elution, the eluted fraction was PCR-amplified with 9 cycles and purified. A 150 bp paired-end reads sequencing was performed on MiSeq personal sequencer (Illumina, San Diego, CA) according to the manufacturer's instructions. 24 pool library samples were loaded on Miseq using MiSeq V3 kit.”

  1. The overall manuscript describes the Illumina method as the NGS method, but PacBio is also one of the NGS methods which may mislead readers.

Dear, reviewer, thank you for your comment. Although they share similarities, to our knowledge PacBio is considered a TGS (third-generation sequencing) method, different from typical NGS method, also called SGS (Second-generation sequencing) methods. We discussed the differences in the introduction part. Below a reference that differenciate PacBio from traditional NGS methods:

https://pubmed.ncbi.nlm.nih.gov/29941292/ van Dijk EL, Jaszczyszyn Y, Naquin D, Thermes C. The Third Revolution in Sequencing Technology. Trends Genet. 2018 Sep;34(9):666-681. doi: 10.1016/j.tig.2018.05.008. Epub 2018 Jun 22. PMID: 29941292.

  1. In Table 1, the total number of cases is 75, but the total number of Femail/male is 77. 

Thank you for you comment. We corrected the number of males in Table 1 and Table 3.

  1. There are 26 family samples, and it is necessary to express whether or not they are family in Table 2 and Table 3.

In Table 1 and Table 3 legends we wrote “The pathology was considered familiar if other cases were present in their clinical familiar history. All the analyzed patients were unrelated.” To exaplain familiarity. We also wrote in “in eight unrelated patients” at line 156 to underline that the probands were all unrelated.

  1. The most important findings of this study appear to be p. (His735_Glu742del). However, this mutation was found in only two samples, and if the two samples belong to one family, the marker is a family-specific mutation, which is likely to be difficult to use in diagnose genetic diseases in the future.

The main topic of the manuscript is to propose PacBio as a new diagnostic method for RPGRORF15 genetic variants sequencing, and the most important findings are the genetic variants that PacBio is able to detect in ORF15 region instead of NGS. Then, considering the occurrence of pathogenic variants in the unstructured ORF15 domain, we modeled the p. (His735_Glu742del) and the interaction between the RPGRORF15 Glu-Gly rich segments and the TTLL5 core domain to derive structural information about the interaction with the observed variants. Molecular dynamics can be useful in increasing our chance to identify pathogenic variants and searching for possible new therapeutic targets.

  1. Although the authors predicted protein structure for p.(His735_Glu742del) mutation, the Table 2 shows the muation was hetero in one sample and homozygotes in the other sample. If the symptoms of the two samples are almost the same, it is necessary to explain whether the Genetic mode of the Mutation is dominant mode or is likely to be influenced by other genes.

The main topic of the manuscript is to propose PacBio as a new diagnostic method for RPGRORF15 genetic variants sequencing. We collected only general clinical data and not specific clinical data that could help us identfying a possible correlation between the state of the genetic variant and the disease state. We added the suggested concept as a limitation of the study (lines 288-289), suggesting that new research could focus in this area.

  1. Supplementary material lacks description and makes it difficult to understand the table or picture.

 We modified the figure legends. Now they should be more understandable.

Considering the current technology and NGS cost, it seems effective to first WGS with Illumina Sequence to find the causative factors at the Genome-wide level, and if you can't find it, it seems effective to sequence with PacBio the area that is difficult to test with Illumina among the candidate genes.

We agree and we wrote this concept in the discussion part, saying that: “long-read sequencing could be a feasible approach to sequence low-complexity regions for diagnostic purposes, considering NGS as a first option for the other regions” (lines 289-291).

Reviewer 3 Report

Comments and Suggestions for Authors

Bonetti et al. used long-read sequencing to identify RPGRORF15 genetic variants. They also performed molecular modeling and dynamics of the RPGR Glu-Gly repeats binding to TTLL5. They showed that long-red sequencing (PacBio in the manuscript) can detect RPGRORF15 variants efficiently and accurately. Molecular dynamics simulations of TTLL5 binding to the RPGRORF15 glutamate-rich segments will help identify pathogenic variants. However, crucial data and method details are missing to support their conclusions. The following issues should be resolved before the manuscript can be considered for publication.

Major comments:

1.     The authors compared the NGS sequencing method with long-read sequencing methods in detecting RPGRORF15 genetic variants. However, crucial methods and sequencing results details are missing to support their conclusions. The following methods and sequencing results are required:

a.      Section 4.2: sequencing library construction method for NGS. Although a kit was used to build the library, it is necessary to summarize the entire procedure, e.g. PCR amplification of the target region, sample multiplexing, number of sequencing run, sequencing cost etc.

b.     Section 4.3: more details for the PacBio sequencing library preparation, e.g. sample multiplexing, number of sequencing run, sequencing cost etc.

c.      Sequencing data analysis methods details.

d.     For NGS analysis, the authors used Illumina Miseq which only yield a few million reads. It is necessary to provide the raw reads statistics for both NGS and TGS to better compare the two methods and support their conclusions. The low coverage of NGS may be caused by limited data output.

Minor comments:

1.     Line 136-139, Table 1: Please explain the term “Familiarity” to make things clear to readers.

2.     Line 177: p.(His735_Glu742del) variant was not mentioned in Table 2. It seems to be variant No.2 or No.3 in table 2. Please add the variant name, p.(His735_Glu742del)  to Table 2 or keep the variant name consistent. 

Author Response

Bonetti et al. used long-read sequencing to identify RPGRORF15 genetic variants. They also performed molecular modeling and dynamics of the RPGR Glu-Gly repeats binding to TTLL5. They showed that long-red sequencing (PacBio in the manuscript) can detect RPGRORF15 variants efficiently and accurately. Molecular dynamics simulations of TTLL5 binding to the RPGRORF15 glutamate-rich segments will help identify pathogenic variants. However, crucial data and method details are missing to support their conclusions. The following issues should be resolved before the manuscript can be considered for publication.

Major comments:

  1. The authors compared the NGS sequencing method with long-read sequencing methods in detecting RPGRORF15genetic variants. However, crucial methods and sequencing results details are missing to support their conclusions. The following methods and sequencing results are required:

  1. Section 4.2: sequencing library construction method for NGS. Although a kit was used to build the library, it is necessary to summarize the entire procedure, e.g. PCR amplification of the target region, sample multiplexing, number of sequencing run, sequencing cost etc.

Dear, in lines 307-335 we added all the relevant information:

“A large custom panel [approximately 2.4 Mb cumulative target length (GRCh38/hg38)], encompassing several genes related to retinitis pigmentosa and cone dystrophy among which RPGR, were used for NGS analysis. The DNA probe set was designed to capture the coding exons and flanking regions of each gene of the panels using the Twist Custom Panel Design Technology (Twist Biosci-ence, https://www.twistbioscience.com/products/ngs). The subpanel of analyzed genes were selected on the basis of literature or databases [Human Gene Mutation Database (HGMD Professional), Online Mendelian Inheritance in Man (OMIM), Orphanet, NCBI GeneReviews, NCBI PubMed and specific database].

Library preparation from genomic DNA samples was performed according to the manufacturer’s protocol using the Twist Library Preparation EF Kit and Twist Universal Adapter (UDI) System with Standard Hybridization Target Enrichment (Twist Bioscience). Briefly for library preparation, 50 ng of each genomic DNA was enzymatically fragmented to yield fragments of 450-550 bp and end repaired and dA-Tailing in the same reaction. Then, the Twist universal adaptor was ligated on fragments, SPRI purified and enriched by 7 PCR cycles with Twist Unique Dual Index Primers. 187.5 ng of each purified Libraries was then hybridized to the Twist oligo probe capture library for 16 hr in a twelve-plex reaction. After hybridization, washing, and elution, the eluted fraction was PCR-amplified with 9 cycles and purified. A 150 bp paired-end reads sequencing was performed on MiSeq personal sequencer (Illumina, San Diego, CA) according to the manufacturer's instructions. 24 pool library samples were loaded on Miseq using MiSeq V3 kit.”

  1. Section 4.3: more details for the PacBio sequencing library preparation, e.g. sample multiplexing, number of sequencing run, sequencing cost etc.

Dear, in lines 336-371 we added all the relevant information:

“The SMRTbell library to sequence RPGRORF15 was prepared following the PacBio protocol guidelines Procedure & Checklist—Preparing SMRTbell® Libraries using Pac-Bio® Barcoded Universal Primers for Multiplexing Amplicons—Part Number 101–791-800 Version 02 (April 2020) [27]. All assays were performed as already described our publica-tion [28].

For the first-round PCR, forward target-specific primer (5'- GACAG-TTACATGGAAGGTGCAA-3') and reverse target-specific primer (5'- TAC-CAG-TGCCTCCTATTGTCTT -3') tailed to F/R universal sequences were designed to am-plify the ~ 1.7 kb DNA fragment spanning the entire RPGRORF15 gene exon 15 sequence (NM_001034853). Primer design was performed using the freely available program Pri-mer3web [29] and, to avoid mispriming, the primer couple was test-ed in silico using NCBI PrimerBLAST [30] and the UCSC tool In-Silico PCR [31].  First- and second-round PCR were performed using PrimeSTAR GXL Polymerase (TaKaRa Bio, Shiga, Japan). The poly-merase was tested and confirmed in our precedent work. Second round PCR products were purifed using 0.45X AMPure PB beads (Pacifc Biosciences) and DNA concentrations were read on a Qubit 2.0 Fluorometer using the Qubit dsDNA Broad Range Assay Kit (Invitrogen Life Technologies, USA).

For an amplicon size ranging from 1 to 3 kb, the input DNA amounts per pool should be between 500 – 1000 ng. Depending on the recorded concentrations, we proceed by cal-culating the necessary microliters of each sample so as to obtain a total amount of 1000 ng. The pooled PCR amplicons were then concentrated using AMPure PB Beads. The next step consists in the construction of the SMRTbell library, which involves DNA damage repair, End-repair/A-tailing and adapter ligation. All the steps are described in the Pacbio proce-dure with specific reagent and amounts [27]. The SMRTbell Templates are then purified again with AMPure PB before sequencing.”

  1. Sequencing data analysis methods details.

Dear, in lines 372-384 we added some relevant information and cited some of our relevant scientific articles published on the bioinformatic analysis of sequencing data (ref. 35,37,39):

“4.4. Bioinformatics and genetic variants classification - The pathogenicity of all identified variants was evaluated according to American College of Medical Genetics and Genomics guidelines (ACMG) [32], with the help of VarSome [33], dbSNP [25], ClinVar [26] and gnomAD [34] databases, thanks to an in-house bioinformatics pipeline [35]. Fastq (forward-reverse) files were obtained after sequencing. Bioinformatic analysis was performed as previously described [36,37]. Briefly, the sequencing reads were mapped to the reference genome (hg38/GRCh38) using Burrow-Wheeler Aligner (version 0.7.17-r1188) software. Duplicates were re-moved using SAMBAMBA (version 0.6.7) and MarkDuplicates GATK (version 4.0.0.0). The BAM alignment files generated were refined by local realignment and base quality score recalibration, using the RealignerTargetCreator and IndelRealigner GATK tools. Minor allele frequencies (MAF) were retrieved from the Genome Aggregation Data-base [38].  Long-read sequencing data were analyzed using an in-house bioinformat-ics pipeline [39].”

  1. For NGS analysis, the authors used Illumina Miseq which only yield a few million reads. It is necessary to provide the raw reads statistics for both NGS and TGS to better compare the two methods and support their conclusions. The low coverage of NGS may be caused by limited data output.

In diagnostics practice, coverage >= 10X with a phredscore >= 18 (phredscore quality called by samtools) has a claimed sensiblity >= 99% (as exaplained in Marceddu, G et al. “Analysis of machine learning algorithms as integrative tools for validation of next generation sequencing data.” European review for medical and pharmacological sciences vol. 23,18 (2019): 8139-8147. doi:10.26355/eurrev_201909_19034 https://www.europeanreview.org/article/19034). Thus, our coverage, which is always higher than 10X apart from in the ORF15 region, does not invalidate result calling. Moreover, considering the coverage of PacBio, we wrote at lines 154,155: “On average 98,12% of reads mapped to the entire genome. Within the region, coverage averaged 3587.44 with a standard deviation of 436.58.”

Minor comments:

  1. Line 136-139, Table 1: Please explain the term “Familiarity” to make things clear to readers.

In Table 1 and Table 3 legends we wrote “The pathology was considered familiar if other cases were present in their clinical familiar history. All the analyzed patients were unrelated.” To exaplain familiarity. We also wrote in “in eight unrelated patients” at line 156 to underline that the probands were all unrelated.

  1. Line 177: p.(His735_Glu742del) variant was not mentioned in Table 2. It seems to be variant No.2 or No.3 in table 2. Please add the variant name, p.(His735_Glu742del)  to Table 2 or keep the variant name consistent. 

Dear, we copied in lines 182,190,236 the corresponding name of the variant as it was reported in Table 2 to keep the variant name consistent.

Round 2

Reviewer 2 Report

Comments and Suggestions for Authors

I think the authors answered my previous questions well, and I have no additional questions.

Reviewer 3 Report

Comments and Suggestions for Authors

The revised version of the manuscript looks better. Bonetti et al. resolved the issues properly. The added information is useful to clarify “Familiarity” and RPGRORF15 genetic variant. The methods details for NGS and long-read sequencing can help us compare the two techniques and encourage more people to try long-read sequencing considering its benefits. I noticed that the NGS libraries were constructed using all the genes including RPGR within a ~2.4 Mb custom genomic region. While the long-read sequencing libraries seemed to be constructed using only RPGRORF15 region. Is it possible to build the NGS libraries only RPGRORF15 region? Will it increase the coverage and sensitivity of identifying genomic variants? Other than that, I have no more concerns.